# Characteristics of Nitrogen Output during Typical Rainfall in Different Sugarcane Growth Stages in a Southern Subtropical Watershed

**Hao Guo** [1]**, Yong Li** [1,*]**, Xu Wang** [1,2]**, Hongyan Ruan** [1,3,*]**, Toyin Peter Abegunrin** [1,4]**, Lanchao Wei** [1]**, Zhigang Huang** [1]**, Kayode Steven Are** [1,5] **and Gabriel Oladele Awe** [6]

[1] State Key Laboratory for Conservation and Utilization of Subtropical Agro-Bioresources, Agricultural College, Guangxi University, Nanning 530004, China; 18269003212@163.com (H.G.); xusange1995@163.com (X.W.); tpabegunrin@lautech.edu.ng (T.P.A.); 1717309002@st.gxu.edu.cn (L.W.); hzg@gxu.edu.cn (Z.H.); kayodeare@gmail.com (K.S.A.)

[2] Scientific Research Academy of Guangxi Environmental Protection, Nanning 530022, China

[3] Key Laboratory of Environment Change and Resources Use in Beibu Gulf, Ministry of Education, Nanning Normal University, Nanning 530001, China

[4] Department of Agricultural Engineering, Ladoke Akintola University of Technology, P.M.B. 4000, Ogbomoso 210214, Nigeria

[5] Institute of Agricultural Research and Training, Obafemi Awolowo University, Moor Plantation, P.M.B. 5029, Ibadan 200254, Nigeria

[6] Department of Soil Resources and Environmental Management, Ekiti State University, P.M.B. 5363, Ado Ekiti 362103, Nigeria; gabriel.awe@eksu.edu.ng

[*] Correspondence: 19968062047@163.com (Y.L.); rhyan@nnnu.edu.cn (H.R.); Tel.: +86-15078836208 (Y.L.); +86-19968062047 (H.R.)

**Abstract:** Excessive fertilizer application, majorly nitrogen- and phosphorus-based fertilizers, in farmland has intensified environmental pollution of rivers, lakes, and other surface water bodies worldwide by agricultural non-point sources, especially the highly-mobile nitrogen. To solve nitrogen pollution in sugarcane areas, exploring the nitrogen output characteristics of agricultural watersheds in crop fields becomes necessary. Therefore, the objective of the study was to evaluate the characteristics of nitrogen output during typical rainfall events in different sugarcane growth stages in a southern tropical watershed in China. Dynamic monitoring of runoff and nitrogen concentration was carried out for four rainfall events and compared among four sugarcane growth stages (Establishment; Vegetative growth; Grand growth; Ripening) during the growing season of 2018 in the Nala watershed, Kelan Reservoir, Guangxi, China. The results showed that the total dissolved nitrogen flux of the 4 rainfall events ranged from 0.08 to 9.88 kg·hm$^{-2}$ for the different growth stages. Nitrate nitrogen was the main component of the total flux, accounting between 75.7 and 92.1% of the total dissolved nitrogen while ammonium nitrogen accounted between 1.80 and 5.26% of the total flux for the 4 rainfall events. Total dissolved nitrogen and nitrate-nitrogen were significantly and negatively correlated with runoff ($p < 0.05$), while total dissolved nitrogen concentration did not correlate with runoff. The incipient scouring effect of total dissolved nitrogen and nitrate-nitrogen was not noticeable. The concentration of total dissolved nitrogen in the Nala watershed was inferior to class V water quality standard, indicating water eutrophication danger. The study showed that nitrogen nutrient inflow into the river was promoted by N-fertilization time and rainfall. Therefore, reasonably reducing N-fertilization dose and post-rain fertilization could effectively reduce nitrogen inflow into rivers and avoid the intensification of eutrophication in sugarcane areas. We recommend multiple years of studies to verify the possible impacts of differences in weather conditions.

**Keywords:** rainstorm; incipient scouring effect; nitrogen fertilization; total nitrogen and nitrate-nitrogen load; environmental pollution

## 1. Introduction

Excessive fertilizer use is a major cause of agricultural non-point source pollution leading to water and environment degradation. Only a part of fertilizer applied to farmland is absorbed and utilized by crops. At the same time, a large proportion of the unused nutrients (nitrogen, N and phosphorus, P) enter water bodies through surface runoff and leaching, causing pollution of surface water bodies [1–3]. The inflow of N and P pollutants into rivers and lakes from farmland is closely related to land use types, rainfall, and vegetation cover, among other factors [4–6]. For example, agricultural activities (crop tillage, planting, management measures) and rainfall can change the watershed's topography, vegetation cover, land use types, and soil nutrient enrichment status. Changes in these factors affect the path and speed of surface runoff confluence, the source of the catchment, and the infiltration and transport process of rainfall runoff, and thus affect the loss of soil nutrients into rivers along with runoff [7,8]. In runoff erosion areas, rainfall is the main driving force of soil nitrogen and phosphorus into water [9]. Characterized by the randomness of the formation process, diversity of influencing factors, wide distribution, and complex monitoring [10], scholars have conducted many studies on the inflow load and source of N and P pollutants from farmland into the rivers [11–17]. In a study conducted in Missouri, USA, Udawatta et al. [11] looked at the total nitrogen inflow load under different crop cover conditions and found that using fertilizer resulted in more nitrogen inflow during periods of heavy rainfall. This finding was corroborated by the findings of [15], who found that nutrients from fertilizers were quickly washed from farmland into rivers by surface runoff during heavy rainfall. Lang et al. [13] detected the concentration of N and P in typical rainfall and runoff from villages in Jiangsu Province, China, and the results showed that the concentration of Dissolved Nitrogen (DN) and Dissolved Phosphorus (DP) in runoff water was 2.57 times and 4.06 times that in rainfalls, respectively, suggesting that DN and DP pollution mainly was due to human activities. Jiang et al. [18] studied the process of nitrogen and phosphorus migration in rainstorm events and found that nitrogen export was dominated by particulate nitrogen in the early stage and nitrate nitrogen in the late stage, while phosphorus was mainly particulate phosphorus. Shao and Zhang [19] studied the interaction between soil and rainfall/runoff in the process of slope runoff on the Loess Plateau and analyzed the interaction and principle between soil nutrients and rainfall/runoff. Huang et al. [20] studied nitrogen loss during rainfall under simulated rainfall conditions and revealed that particulate nitrogen was the primary form of nitrogen loss in rainstorm runoff, and erosion sediment had an enrichment effect. However, these studies only considered the impact of vegetation cover, rainfall, and other factors on nitrogen pollutants in the river. They paid little attention to the nutrient output corresponding to land cover changes at different crop growth stages. Therefore, at the watershed scale, especially in karst areas, it is essential to determine the characteristics of nitrogen output during rainfall at different growth stages of crops.

Sugarcane is a necessary sugar and energy cash crop in the world. China is the 4th largest sugarcane producer globally, accounting for 6.05% of the world's sugarcane cultivation area and 6.2% of the total sugarcane output [9]. Sugarcane plays a crucial role in China's agricultural and industrial economy because of its renewable supply of sugar and by-products such as biofuels and fiber [21,22]. Located in the subtropical region, Guangxi has the largest sugarcane planting area in China. Its planting area accounts for more than 85% of the sugarcane planting area in China, and its sugar production accounts for more than 90% of the total sugar production in China [21]. Sugarcane is a high-yield crop containing many plant nutrients; every 100 mg straw needs to accumulate 100–154 kg N, 15–25 kg $P_2O_5$, 77 kg $K_2O$, and 14–49 kg S [23]. A lot of nutrients need to be absorbed from the soil during the growth period. At present, the sustainable production of sugarcane is achieved mainly through massive fertilization to ensure that soil fertility does not degrade [24]. The sugar content and sugarcane yield will decrease if fertilization is insufficient during growth. Excessive fertilization (especially nitrogen fertilizer) will not only prolong sugarcane's growth period but also decrease the yield and quality of sucrose

and increase planting costs, greenhouse gas emissions, and groundwater pollution [25]. In recent years, with the continuous increase in the use of chemical fertilizer and the sub-tropical monsoon climate of the Guangxi region (characterized by short duration and intense rainfall pattern), the problem of agricultural non-point source pollution in this region has become very prominent. Understanding the dynamics and monitoring of runoff, analysis of change in N concentration and its output characteristics in small/medium-sized watersheds under different rainfall events could provide a scientific basis for preventing and controlling agricultural non-point source pollution in an area. We hypothesized that N flux into a river during rainfall events cum N-fertilization will differ during different growth stages of sugarcane. Therefore, in this study, a typical small intensive agricultural watershed (Nala watershed) in Guangxi was selected to monitor and explore the characteristics of N inflow into the river during the four stages of sugarcane growth (Establishment, Tillering, Grand Growth, and Ripening) under rainstorm conditions.

## 2. Materials and Methods

### 2.1. Study Area

The study was conducted in Nala, a small watershed upstream of Kelan Reservoir, Dongluo Town, Fussui, Chongzuo, Guangxi. The area of the watershed is 1.28 km$^2$ and is located between longitude 107°39′29″~107°40′17″ E and latitude 22°20′36″~22°20′36″ N in the south of the Tropic of Cancer. The study area has a subtropical monsoon climate characterized by abundant rainfall (about 1400 mm per year). The annual sunshine duration is about 1600 h, with an annual average temperature ranging between 20.8 °C and 22.4 °C. The rainfall is generally concentrated between March to September, with uneven distribution, and occurs between 30 and 200 days per year [26]. Nala watershed has a sufficient photoperiod throughout the year. With both sunshine and rainfall occurring during the same season, the watershed has been suitable for producing sugarcane and other cash crops [27]. The area's terrain is dominated by foothills with gradients ranging from 0° to 35°, and the mean altitude of the area is about 1200 m a.s.l.

The soil type in the area belongs to the taxonomy class is latosolic, a predominant reddish lateritic soil with silty loam topsoil texture [12]. The average infiltration rate for the soil is 6.98 mm·min$^{-1}$. The watershed's boundary is mainly surrounded by economic forests, with Eucalyptus (*Eucalyptus robusta* S.) as the main economic forest and sugarcane (*Saccharum officinarum* L.) as the main cash crops. Sugarcane, planted mainly on the hillslope, accounted for about 74% of the total land use in the watershed, while an insignificant portion of the hillslope (with a gentle gradient) has been cropped with rice. In 2018, the sugarcane variety in the Nala watershed was Guitang-42. The planting row spacing was 1 m, and the planting depth was 25 cm. The soil was cultivated during planting but not at a later stage. Due to late planting time and rising temperature, sugarcane is not coated when planting. Since 1980, the fertilizer used in the watershed has been majorly inorganic and is usually applied twice as topdressing. The reason was that sugarcane was an annual grass plant, and its whole growth stage could be divided into seedling, tillering, elongation, and maturity. Sugarcane had different nutrient demands at different growth stages, so sugarcane planting applied base fertilizer and elongation period topdressing. In 2018, an annual total of 296.4 kg·ha$^{-1}$ N and 22.9 kg·ha$^{-1}$ P fertilizers were applied (by broadcasting method) for sugarcane production in the entire watershed. The first application of 205.3 kg·ha$^{-1}$ N and 15.9 kg·ha$^{-1}$ P fertilizers are executed between June and July, while the second application of 91.1 kg·ha$^{-1}$ N and 7.0 kg·ha$^{-1}$ P fertilizers are carried out in mid-late August. At the crop establishment stage, farmers replant and re-fertilize plants, depending on the emergence of seedlings in different areas of the watershed.

### 2.2. Field Monitoring and Sample Collection and Analysis

2.2.1. Flow and Rainfall Monitoring

A monitoring station (Figure 1) was set at the egress section of the watershed. A Bacher trough and an automatic water and sediment sampler (ISCO6712, Teledyne, USA)

were installed at the egress. These were used to monitor the change in water level and flow in real-time online. Real-time rainfall data were obtained by rain gauges (three) installed within the watershed experimental stations. In this study, the water and sediment samples were collected at the outlet (S1, Figure 1) monitoring station of the watershed, representing the overall situation. A total of four rainfall measurement campaigns, one for each of the sugarcane growth stage, were collected as follow: Establishment (June), Vegetative growth (July), Grand growth (September), and Ripening (November).

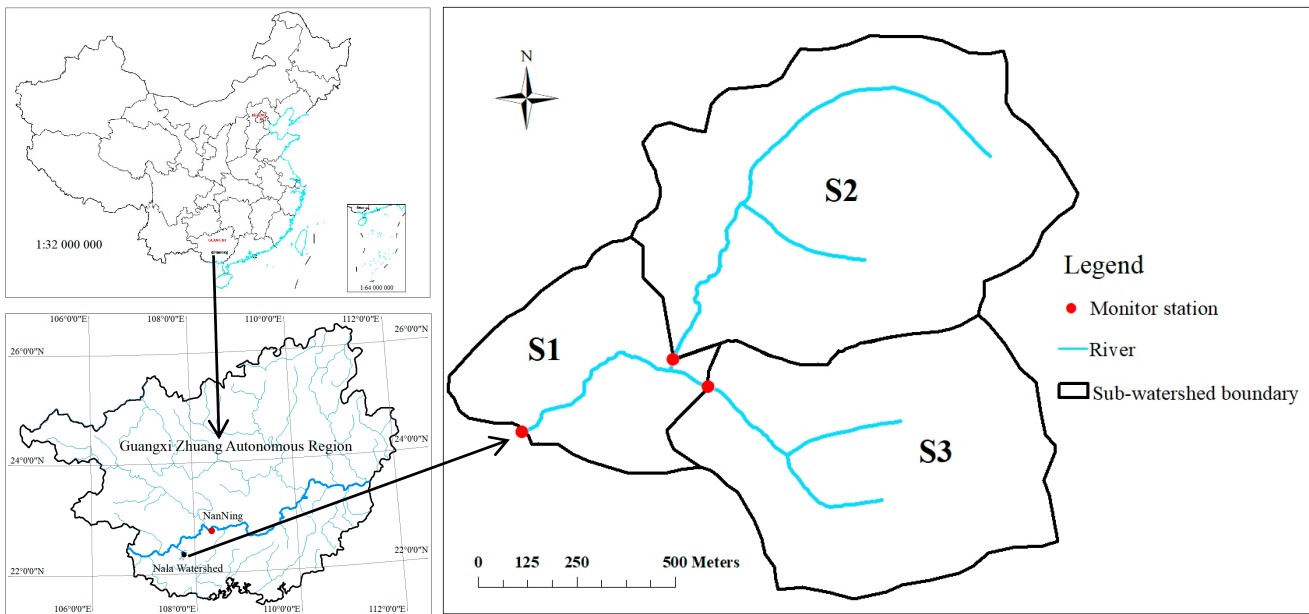

**Figure 1.** Sampling points in the Nala watershed.

### 2.2.2. Water and Sediment Sample Collection

The automatic water and sand sampler (ISCO6712) was installed and set according to the description [9]. At a water level 2 cm higher than normal conditions, the sampler will delay for 1 min and start to collect No. 1 sample, then collect No. 2–4 samples (3 samples) at an interval of 15 min; No. 5–8 samples at an interval of 30 min and No. 9–12 samples (4 samples) at an interval of 60 min. Samples No. 13–16 (4 samples) were collected at 120 min, and No. 17–24 samples (7 samples) were collected at 240 min. The ISCO6712 sediment sampler stopped sampling when the water level in the tank was less than 2 cm higher than the normal water level. The total number of samples collected was 24 (1–24 bottles). The collected mixture of sediment and water was transferred to 1 L sealed plastic bottles, marked with sampling time and place, and transported in cooled containers to the laboratory. In the laboratory, all samples were immediately placed in a refrigerator set at 4 °C, and all analyses were carried out within 24 h of collection.

### 2.3. Laboratory Analysis

The water samples were filtered using a 0.45 μm filtering membrane for the determination of total dissolved nitrogen (TDN), nitrate-nitrogen ($NO_3^-$-N), and ammonium nitrogen ($NH_4^+$-N). Dissolved N was determined by Potassium persulfate oxidation and ultraviolet spectrophotometry (GB11894-89). $NH_4^+$-N was determined by using Nessler's reagent colorimetry (HJ 535-2009). $NO_3^-$-N was determined by ion chromatography (HJ 84-2016) [9].

### 2.4. Data Calculation and Analysis

2.4.1. Calculation of Runoff and Nitrogen Load

Runoff and the nitrogen loss load were calculated according to Equations (1) and (2), respectively [28].

$$V = \int_0^t q_t(t)dt = \sum_{i=1}^n Q_i \Delta t_i \tag{1}$$

where $V$ represents single rainfall-runoff (m$^3$), $Q_i$ is the instantaneous Flow during monitoring sample $i$ (m$^3 \cdot$h$^{-1}$), and $\Delta t_i$ is the sampling time interval between sample $i$ and sample $i-1$ (h).

$$M = \int_0^t c_t \times q_t(t)dt = \sum_{i=1}^n c_i Q_i \Delta t_i \tag{2}$$

where $M$ is the runoff nutrient output load of a single rainfall event ($q$), $c_i$ is the nutrient concentration of sample $i$ in the runoff (mg$\cdot$L$^{-1}$), $\Delta t_i$ is the sampling time interval between sample $i$ and sample $i-1$(s), and $Q_i$ is the instantaneous Flow during monitoring the sample $i$ (m$^3 \cdot$s$^{-1}$).

2.4.2. Calculation of Event Mean Concentration (*EMC*)

The Event Mean Concentration (*EMC*) is used to evaluate the pollution degree of rainfall events. The *EMC* is calculated using Equation (3) as described in [29,30].

$$EMC = \frac{M}{V} = \frac{\sum C_t Q_t \, \Delta t}{\sum Q_t \, \Delta t} \tag{3}$$

In the above formula, *EMC* is the rainfall event mean concentration, $M$ and $V$ are as defined in Equations (1) and (2), respectively, $C_t$ is the nutrient concentration of the sample (mg$\cdot$L$^{-1}$), $Q_t$ is the runoff of the period (m$^3 \cdot$s$^{-1}$), and $\Delta t$ is as defined in the Equations above. Calculation of cumulative pollutant load curve.

The cumulative pollutant load curve was used to evaluate the incipient scouring effect of rainfall events. The cumulative pollutant load curve was calculated according to Equation (4) [31].

$$\beta = \frac{\sum C_t Q_t \, \Delta t / M}{\sum Q_t \, \Delta t / V} \tag{4}$$

where $\beta$ is the cumulative rate of pollution load and $C_t$, $Q_t$, $\Delta t$, $M$, $V$ already defined in Equation (3) above.

When the load accumulation curve $\beta > 1$ and the slope is greater than 45°, it indicates that pollutants have an initial scouring effect [31].

### 2.5. Statistical Analysis

All data were processed and analyzed using Excel 2019 and SPSS version 21.0. Loads of exported dissolved nutrients were subjected to descriptive statistics to determine the mean, minimum, and maximum values. Analysis of variance (ANOVA) was carried out on the nutrients export for the test of significance among the different crop growth stages using randomized complete block design while means were separated using Fisher's Least Significant Difference (LSD) test at $p \le 0.05$ level of significance or as otherwise indicated. Regression analyses were performed to establish the relationships between concentrations and exports of dissolved nutrients at different growth stages and rainfall and runoff.

## 3. Results

### 3.1. Nitrogen Output during Different Rainfall Events

The characteristics of rainfall and runoff during the different sugarcane growth stages are presented in Table 1. The rainfall depth, runoff volume, and intensity were maximum during the grand growth stage. The rainfall at grand growth was about 2–3 times greater than the rainfall depth at other growth stages; the total runoff volume was 2–30 times, and

the mean rainfall intensity was twice the intensity at other growth stages. The rainfall depth at grand growth was significantly higher ($p < 0.05$) than other growth stages. However, there was no significant difference ($p > 0.05$) in the rainfall depths during vegetative growth and ripening stages. The runoff volumes at all the growth stages differed significantly ($p < 0.05$). The mean intensity of the rainfall during the grand growth stage was significantly higher ($p < 0.05$) than at other growth stages. The rainfall depth was in the order: of grand growth > ripening > vegetative growth > establishment, while the total runoff was: grand growth > vegetative growth > ripening > establishment, and the mean rainfall intensity was: grand growth > establishment > ripening > vegetative growth.

**Table 1.** The characteristics of rainfall events at different sugarcane growth stages.

| Rainfall Events | Rainfall (mm) | Total Runoff (m$^3$) | Runoff Duration (h) | Rainfall Duration (h) | Mean Rainfall Intensity (mm·h$^{-1}$) |
|---|---|---|---|---|---|
| Establishment | 29.2 | 11,333 | 14.8 | 4.95 | 5.89 |
| Vegetative growth | 40.2 | 40,598 | 18.8 | 9.55 | 4.21 |
| Grand growth | 77.6 | 91,589 | 34.8 | 7.25 | 10.71 |
| Ripening | 41.4 | 2981 | 12.8 | 7.49 | 5.53 |

Figure 2 shows the changes in runoff volume and dissolved nitrogen concentration during different rainfall events. The duration of the runoff process was in order: grand growth > vegetative growth > establishment > ripening, which was 2086, 1126, 886, and 766 min, respectively. The average TDN concentration (22.20 mg·L$^{-1}$) was highest during the establishment stage and reached 13.0 mg·L$^{-1}$ at the peak flow (2870.496 m$^3$·h$^{-1}$). At the end of the rainfall event, NO$_3^-$-N concentration followed a similar trend as TDN's. The fluctuation in the degree of NH$_4^+$-N concentration was slight. The runoff during the vegetative growth stage had 2 peak values during the whole rainfall period; the runoff reached the 1st peak value (1436.272 m$^3$·h$^{-1}$) after the beginning of the rainfall event and then decreased rapidly, the runoff appeared at the 2nd peak value (11,236.595 m$^3$·h$^{-1}$) at the later stage of the rainfall event. TDN and NO$_3^-$-N showed a sinusoidal trend, decreasing and increasing, while NH$_4^+$-N increased first and then decreased. During rainfall events in the Grand growth stage, the TDN showed a decrease-increase-decrease trend, while the concentration of NH$_4^+$-N did not change significantly. The lowest TDN (8.64 mg·L$^{-1}$) appeared at the flow peak (21,845.181 m$^3$·h$^{-1}$). In comparison, the maximum value (18.53 mg·L$^{-1}$) appeared with the minimum runoff, which was because NH$_4^+$-N fertilizer was applied by farmers in late August and entered the water with runoff during rainfall scouring, resulting in the overall increase in NH$_4^+$-N concentration in water.

During the Ripening stage, the runoff of rainfall events had little change, and the concentrations of TDN, NH$_4^+$-N, and NO$_3^-$-N were low. The TDN first decreased, later increased, and then basically remained unchanged. The variation of TDN ranged from 1.28 mg·L$^{-1}$ to 3.79 mg·L$^{-1}$. The concentration of NH$_4^+$-N varied slightly, ranging from 0.091 mg·L$^{-1}$ to 0.432 mg·L$^{-1}$. It was presumed that the ripening stage happened during the dry season when the soil surface was covered with dead leaves, protecting from rainfall causing less runoff and sediment into the river.

Figure 3 shows the results of Pearson correlation analysis of nitrogen and runoff in the 3 rainfall events. The 3 rainfall events showed a significant negative correlation between runoff and TDN concentration ($p < 0.05$). There was no significant correlation between NH$_4^+$-N concentration and runoff. A significant positive correlation was found between NO$_3^-$-N and TDN concentrations in three rainfall events.

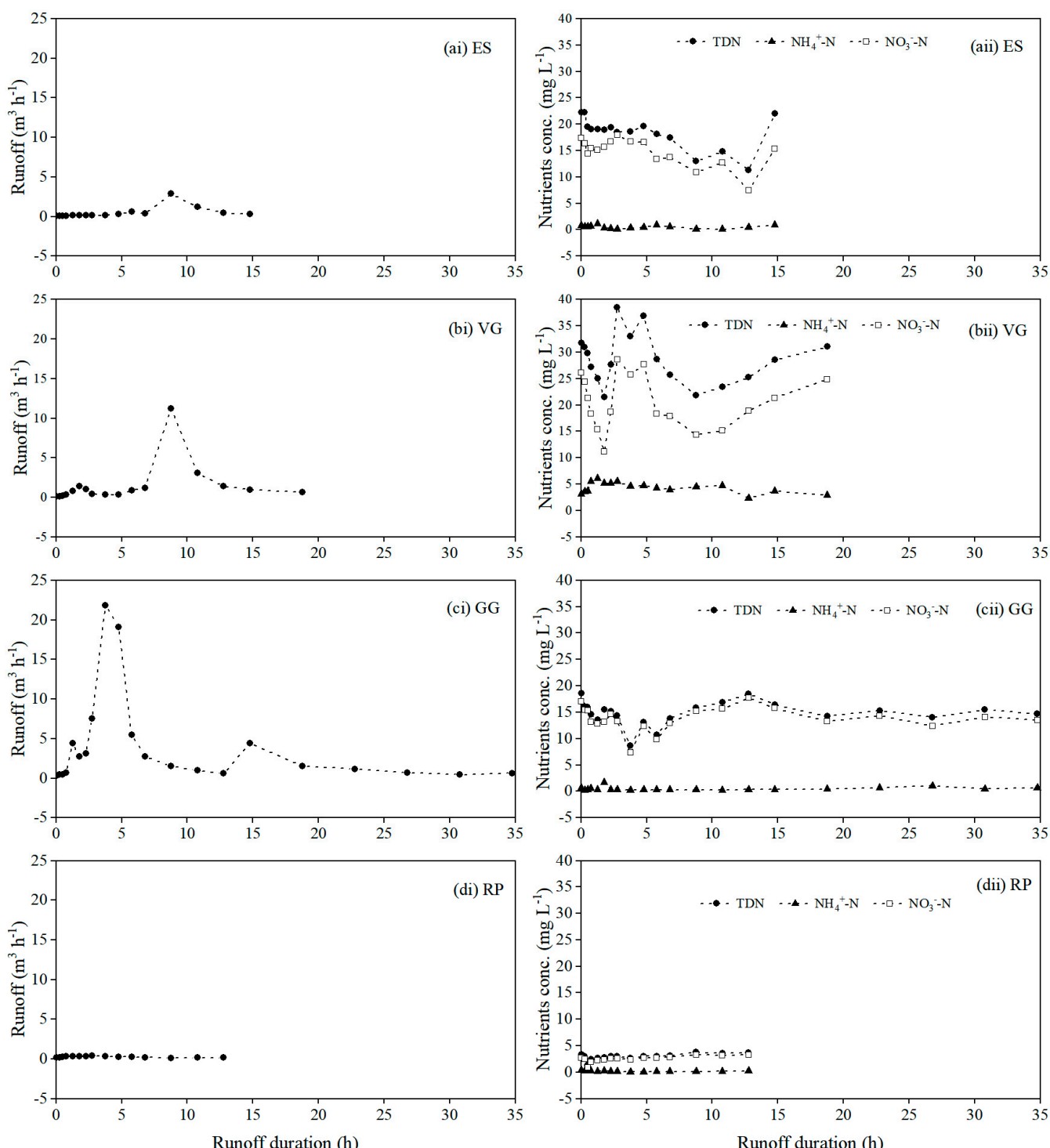

**Figure 2.** The changes of runoff (**i**) and nitrogen concentration (**ii**) in different rainfall processes during the sugarcane growth stages in (**a**) Establishment (ES), (**b**) Vegetative growth (VG), (**c**) Grand growth (GG), and (**d**) Ripening (RP).

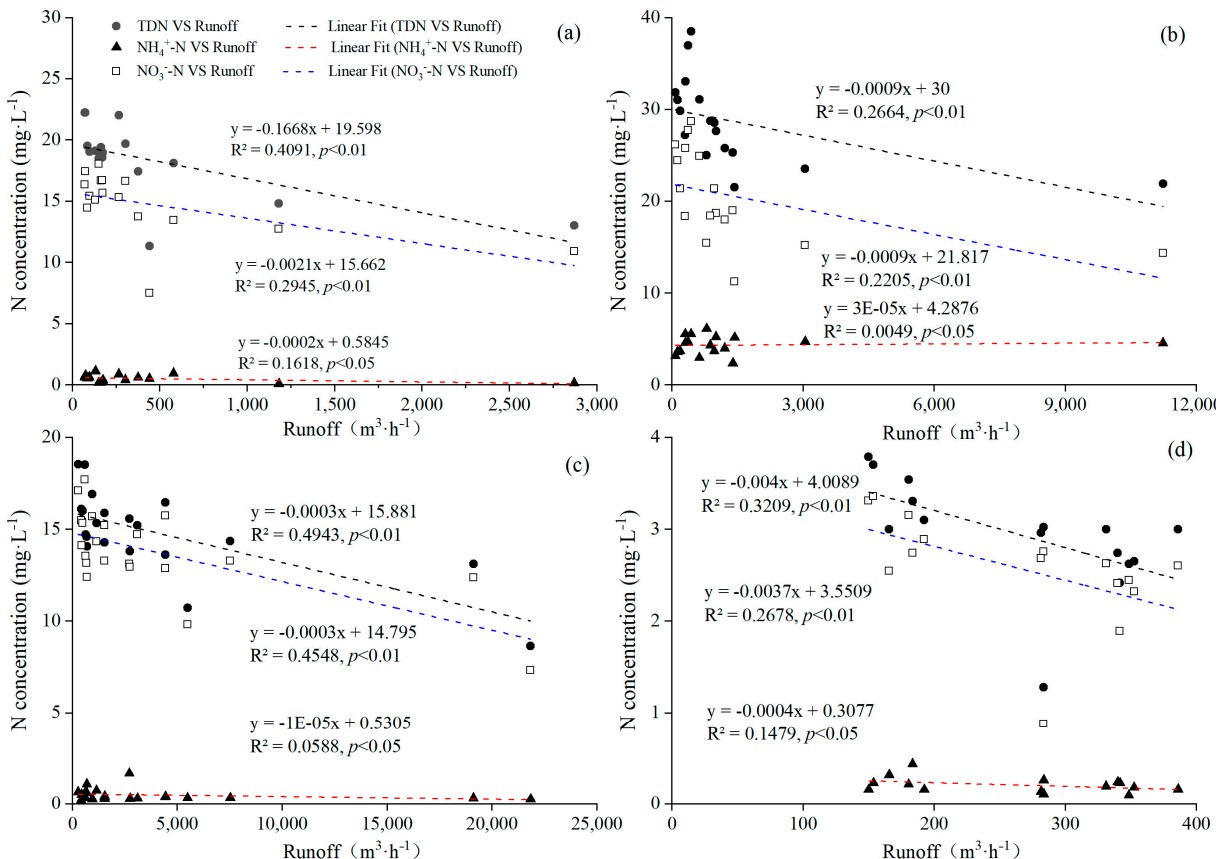

**Figure 3.** Correlation between total dissolved nitrogen (TDN), ammonia nitrogen ($NH_4^+$-N), nitrate nitrogen ($NO_3^-$-N), and runoff during the sugarcane growth stages in (**a**) Establishment, ES, (**b**) Vegetative growth, VG (**c**) Grand growth, GG and (**d**) Ripening, Rp.

### 3.2. EMC in Different Rainfall Events of Different Forms of Nitrogen

*EMC* is an indication of the pollution degree of the whole rainfall. Table 2 shows the TDN, $NO_3^-$-N, and $NH_4^+$-N *EMC* for the 4 rainfall events. The *EMC* of TDN ranged from 3.08 mg·L$^{-1}$ at the Ripening stage to 23.93 mg·L$^{-1}$ at the vegetative growth stage. The *EMC* of TDN at the vegetative growth stage is significantly ($p < 0.05$) the highest, being 2–8 times the *EMC* at other growth stages, and it was in the order: Vegetative growth > Establishment > Grand growth > Ripening. Similar trends were observed for the *EMC* of $NH_4^+$-N and $NO_3^-$-N, with the maximum and minimum values observed at Vegetative growth and Ripening stages. However, the values for $NH_4^+$-N were generally low.

**Table 2.** Output characteristics of nitrogen concentration (mg·L$^{-1}$) in different rainfall events.

| Rainfall Events | TDN (mg·L$^{-1}$) | | | AN (mg·L$^{-1}$) | | | NN (mg·L$^{-1}$) | | |
|---|---|---|---|---|---|---|---|---|---|
| | Max | Min | *EMC* | Max | Min | *EMC* | Max | Min | *EMC* |
| Establishment | 22.2 | 11.3 | 14.54 | 1.1 | 0.07 | 0.26 | 18.01 | 7.47 | 11.86 |
| Vegetative growth | 38.5 | 21.5 | 23.93 | 6.1 | 2.34 | 4.29 | 28.67 | 11.2 | 16.28 |
| Grand growth | 18.53 | 8.64 | 12.94 | 1.65 | 0.19 | 0.38 | 17.69 | 7.31 | 11.92 |
| Ripening | 3.79 | 1.28 | 3.08 | 0.43 | 0.09 | 0.17 | 3.35 | 0.87 | 2.75 |

TDN: Total Dissolved Nitrogen; AN: Ammonium nitrogen ($NH_4^+$-N); NN: Nitrate nitrogen ($NO_3^-$-N); *EMC*: Event Mean Concentration; Max: Maximum; Min: Minimum.

### 3.3. Nitrogen Flux Output during Different Rainfall Events

The nitrogen flux outputs during the growth stages are presented in Table 3. The TDN flux ranged from 9.878 kg·hm$^{-2}$ at Grand growth to 0.076 kg·hm$^2$ at ripening. A

similar trend was observed for the $NO_3^-$-N flux, having the same order Vegetative growth > Grand growth > Establishment > Ripening (Table 3). The output flux of $NH_4^+$-N was maximum at the Vegetative growth stage and minimum at the Ripening stage. $NO_3^-$-N is the major form of dissolved nitrogen flux, accounting for 92% of the total TDN flux output.

**Table 3.** Characteristics of nitrogen output flux in different rainfall events.

| Rainfall Events | Nitrogen Flux, kg·hm$^{-2}$ | | | Output Ratio, % | |
|---|---|---|---|---|---|
| | TDN | AN | NN | AN/TDN | NN/TDN |
| Establishment | 1.373 | 0.025 | 1.120 | 1.80 | 81.52 |
| Vegetative growth | 8.094 | 1.451 | 5.508 | 17.92 | 75.66 |
| Grand growth | 9.878 | 0.287 | 9.097 | 2.91 | 92.09 |
| Ripening | 0.076 | 0.004 | 0.068 | 5.26 | 89.47 |

TDN: Total Dissolved Nitrogen; AN: Ammonium nitrogen ($NH_4^+$-N); NN: Nitrate nitrogen ($NO_3^-$-N).

*3.4. The Cumulative Nitrogen Load Curve*

The cumulative pollutant load curve was used to evaluate the incipient scouring effect of rainfall events [32,33]. The load accumulation curve ratio > 1 indicates that the pollutants have an initial scouring effect. The cumulative pollutant load curve of different forms of nitrogen in the 4 rainfall events is shown in Figure 4. At the crop Establishment stage, the cumulative load curves of TDN and $NO_3^-$-N overlapped, indicating that the cumulative loads were similar. The initial scouring effect was not noticeable. There was an initial solid scouring effect of $NH_4^+$-N in the early rainfall stage, and then it gradually weakened. At the Vegetative growth stage, $NH_4^+$-N had an initial scouring effect at the early rainfall stage, whereas this did not happen to TDN and $NO_3^-$-N. TDN, $NH_4^+$-N, and $NO_3^-$-N produced an initial scouring effect at the Grand growth stage. At the crop Ripening stage, the initial scouring effect of $NH_4^+$-N was evident in the early stage, while that of TDN and $NO_3^-$-N did not occur.

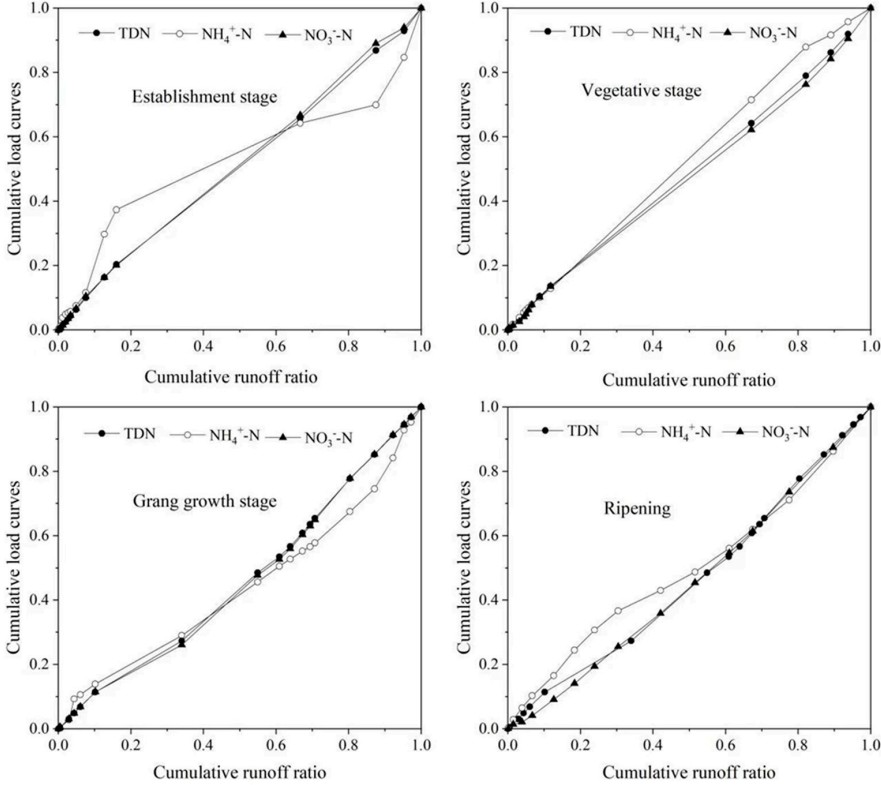

**Figure 4.** Cumulative load curves for rainfall events of different sugarcane grow stages.

## 4. Discussion

### 4.1. Variation in Characteristics of Nitrogen Concentration

In the four rainfall events at different sugarcane growth stages, the trend of TDN and $NO_3^-$-N was the same, which proved that $NO_3^-$-N was the main component of TDN [28].

The long duration of low-intensity rainfall events at the Establishment stage in the early stage was small, causing a relatively high concentration of TDN and $NO^{3-}$-N [34], showing that TDN was the main nitrogen form in runoff when rain intensity was low. In June, the highest temperature in the Nala watershed was 38 °C. High temperature and drought cause soil organic nitrogen mineralization into inorganic nitrogen. When the rainfall occurred, a large amount of nitrate nitrogen entered the water through runoff [35]; as a result, the TDN concentration was the highest at the beginning of the rainfall event at the establishment. With frequent rainfall events, large amounts of runoff dilute the concentration of the pollutant; hence, the minimum concentration of TDN was obtained during the runoff peak. After the rainfall stopped, the TDN and $NO_3^-$-N showed an apparent rising trend because the subsurface flow was dominant in the late rainfall stage, and the subsurface flow was abundant and carried more $NO^{3-}$-N [18]. Therefore, the TDN and $NO_3^-$-N showed an apparent upward trend in the late rainfall. Rainfall at Vegetative growth and Grand growth belonged to short-time heavy rainfall. The runoff peaked within one hour after the occurrence of rainfall and then decreased rapidly. The concentration of TDN and $NO_3^-$-N decreased with the runoff increase, and now the runoff had an obvious dilution effect.

At Vegetative growth, there was peak flow in the middle-late period of rainfall; the concentration of TDN and $NO_3^-$-N subsequently decreased and then had an upward trend. It was because the rainfall increased in the late period of rainfall and then a large amount of runoff diluting the concentration of pollutants, resulting in a decrease in the concentration of TDN and $NO_3^-$-N in water.

At Grand growth, the concentrations of TDN and $NO_3^-$-N increased first and then decreased in the late period of rainfall. The reason was that in the late stage of rainfall, the concentrations of TDN and $NO_3^-$-N in water increased due to subsurface flow. In mid-late August, 91.1 $kg \cdot ha^{-1}$ N were applied and after fertilization, part of the fertilizer was absorbed by sugarcane, and part of the fertilizer entered the river with rainfall runoff. Therefore, the concentration of total nitrogen and nitrate nitrogen was high in the pre-rainfall period of the sugarcane growth period. The amount of fertilizer applied and the time of fertilization affected the flow of dissolved nutrients into the river [12]. However, with the loss of subsurface Flow and no subsurface flow into the runoff, the concentration of TDN and $NO_3^-$-N in water decreased rapidly.

During the rainfall event at Ripening, TDN, $NH_4^+$-N, and $NO_3^-$-N did not change significantly. The ripening stage was in November, which corresponds to the dry season, with rainfall producing little runoff, which had a low impact on the nutrients in the surface soil. Therefore, the concentrations of TDN, $NH_4^+$-N, and $NO_3^-$-N were basically unchanged during the period.

The nitrogen concentration and nitrogen load of four rainfall events were compared. The study of Liu et al. [36] revealed that the flow decreased with an increase in coverage, and the effect of runoff interception was more prominent. Zhang et al. [37] found that $NH_4^+$-N and $NO_3^-$-N were mainly lost with runoff. When sugarcane was in the Grand growth, the coverage was high, so although the average rainfall intensity of the rainfall event was high, the TDN concentration and nitrogen losses were less than those at the Vegetative growth [38], showing that compared with rainfall intensity, one-time fertilization significantly increased nitrogen loss. Liang et al. [39] showed that under low rainfall intensity, fertilization at short intervals would still cause much nitrogen loss. The number of dry days before rainfall, rainfall intensity, coverage, fertilization, and other factors caused the variation of nitrogen concentration and nitrogen output load; the order was as follows: Grand growth > Vegetative growth > Establishment > Ripening.

The correlation between the runoff of the four rainfall events and the concentration of different forms of nitrogen was analyzed. The result is shown in Figure 3. In the four rainfall events, the concentration of TDN and $NO_3^--N$ showed a significant negative correlation with runoff, which is consistent with the study of Gao et al. [40]. There was no correlation between $NH_4^+-N$ and runoff.

Taking TDN concentration as the evaluation index, all samples in 3 rainfall events exceeded the Class V standard of the national Surface Water Environmental Quality Standard (GB3838-2002) (2 mg·$L^{-1}$), which was inferior to class V water quality, indicating severe nitrogen pollution in the small watershed. Therefore, reasonably adjusting fertilization dose and post-rain application could effectively reduce nitrogen fertilizer inflow into the river and avoid the intensification of eutrophication and nitrogen pollution in sugarcane-producing areas.

*4.2. Incipient Scouring Effect of Different Rainfall Events*

The incipient scouring effect refers to the fact that most of the pollutants produced by rainfall are lost at the beginning of rainfall. Pollutant concentrations are usually highest at the beginning of rainfall [41]. The incipient scouring effect is related to the types of pollutants with different incipient scouring effects. In this paper, the initial scouring effect of $NH_4^+-N$ was more evident than that of $NO_3^--N$ and TDN, which is consistent with the research results of Lang et al. [13]. The study of Zhang et al. [42] in Lanlingxi Watershed proved that the initial scour effect of $NH_4^+-N$ was evident, not $NO^{3-}-N$. Other studies have pointed out that the ratio of the cumulative load curve was related to the impervious underlying surface area of the watershed [43]. The higher the impervious underlying surface area, the lower the pollutant cumulative load curve ratio will be. In general, the incipient scouring effect of pollutants is affected by many factors.

**5. Conclusions**

Nitrate nitrogen ($NO_3^--N$) was the main component of the TDN flux, while ammonium nitrogen ($NH_4^+-N$) had less output during the different growth stages.

TDN and $NO_3^--N$ were negatively and significantly correlated with runoff ($p < 0.05$), while $NH_4^+-N$ concentration was not significantly correlated.

The average *EMC* of the four rainfall events was lowest at ripening and highest at the vegetative growth stage.

The concentration of TDN in the Nala watershed exceeded class V of the national surface water environmental quality standard, which is inferior to Class V water quality, showing a hidden danger of water eutrophication.

The cumulative nitrogen load curve showed that the incipient scouring effect of $NH_4^+-N$ was obvious at the early period of rainfall, but not at the late period. The incipient scouring effect of TDN and $NO_3^--N$ was not evident.

Nitrogen nutrient inflow into the river was strongly affected by fertilization time and rainfall.

Therefore, reasonably adjusting fertilization dose and post-rain fertilization could effectively reduce nitrogen fertilizer inflow into the river and avoid the intensification of eutrophication and nitrogen pollution in sugarcane areas.

Furthermore, we recommend multiple years of studies to verify the possible impacts of differences in weather conditions.

**Author Contributions:** Conceptualization, H.G. and H.R.; methodology, H.G.; software, X.W.; validation, H.G., H.R. and Z.H.; formal analysis, Z.H.; investigation, X.W.; resources, Z.H.; data curation, L.W.; writing—original draft preparation, H.G.; writing—review and editing, Y.L., G.O.A. and T.P.A.; visualization, K.S.A.; supervision, H.R. and Y.L.; project administration, H.R.; funding acquisition, Z.H. All authors have read and agreed to the published version of the manuscript.

**Funding:** This work was supported by the National Natural Science Foundation of China Grant Number: 42220104004). The Open Project of the State Key Laboratory for Conservation and Utilization of Subtropical Agro-Bioresources funded the study (SKlCUSA-B01905).

**Institutional Review Board Statement:** Not applicable.

**Data Availability Statement:** Data will be made available on request.

**Acknowledgments:** Thanks to Ximin Xu and Mangxia Wei for supporting this paper.

**Conflicts of Interest:** The authors declare no conflict of interest.

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
