# Peer review of "Characteristics of Nitrogen Output during Typical Rainfall in Different Sugarcane Growth Stages in a Southern Subtropical Watershed"

_agriculture, doi:10.3390/agriculture13081613_

Round 1

Reviewer 1 Report

The work entitled: "Characteristics of nitrogen output during typical rainfall in different sugarcane growth stages in a southern subtropical watershed" presents an interesting aspect to explore the nitrogen output characteristics of agricultural watershed in sugarcane areas, in order to solve the nitrogen pollution in sugarcane area. In addition, the study is providing a useful methodology and important reference data for future research. I recommend some minor errors which should be considered.

1) Abstract should present a clear purpose and scope of the work, provide a short research methodology and present the most important results and conclusions. It needs to be summarized.

2) The Keywords should be more concise.

3) In the introduction, the authors should put forward an alternative research hypothesis, in relation to the null hypothesis. Verification of the hypothesis should be carried out later in the work.

4) There are a large number of typo-errors.

5) The conclusion should be more concise, generalizing and summarizing.

6) The use of literature should be checked carefully.  

7) English editing is highly recommended.

English editing is recommended.

Author Response

We thank this Reviewer for his valuable to review our paper. We agree with the observations and comments to improve the work. The responses to the comments are highlighted below:

Comments and Suggestions for Authors

The work entitled: "Characteristics of nitrogen output during typical rainfall in different sugarcane growth stages in a southern subtropical watershed" presents an interesting aspect to explore the nitrogen output characteristics of agricultural watershed in sugarcane areas, in order to solve the nitrogen pollution in sugarcane area. In addition, the study is providing a useful methodology and important reference data for future research. I recommend some minor errors which should be considered.

1) Abstract should present a clear purpose and scope of the work, provide a short research methodology and present the most important results and conclusions. It needs to be summarized.

Response: The Abstract has been revised and all the key issues raised have been addressed

2) The Keywords should be more concise.

Response: The keywords have been revised as suggested

3) In the introduction, the authors should put forward an alternative research hypothesis, in relation to the null hypothesis. Verification of the hypothesis should be carried out later in the work.

Response: We an alternative research hypothesis in the study, as observed

4) There are a large number of typo-errors.

Response: We have used Grammarly Premium software to edit and proofread the text. We have also given to a senior colleague who is vast in editing and proofreading to proofread.

5) The conclusion should be more concise, generalizing and summarizing.

Response: The Conclusion has been revised, with the key findings summarized

6) The use of literature should be checked carefully.  

Response: We have reviewed the citing and listing of literature used in the text. A sample paper from the Journal page was downloaded and followed.

7) English editing is highly recommended.

Response: We have used Grammarly Premium software to edit and proofread the text. We have also given to a senior colleague who is vast in editing and proofreading to proofread.

Comments on the Quality of English Language

English editing is recommended.

Response: We have used Grammarly Premium software to edit and proofread the text. We have also given to a senior colleague who is vast in editing and proofreading to proofread.

Reviewer 2 Report

"The soil type in the area is predominantly reddish lateritic soil with a silty loam topsoil texture". ( what is the Taxonomic classification for this soil?)

It would be beneficial to show the infiltration rate for this soil. 

In the statement below you are suggesting the adjustment of fertilization timing - can you be specific as to timing for practical purposes?

"Because fertilization before rainfall produces serious agricultural non-point source pollution, adjusting fertilization time reasonably could be an effective measure to avoid rainfall scouring after fertilization and then reduce nitrogen pollution in sugarcane areas."

 Some grammar/editorial corrections are needed throughout the manuscripts. 

Author Response

We thank this Reviewer for his valuable to review our paper. We agree with the observations and comments to improve the work. The responses to the comments are highlighted below:

Comments and Suggestions for Authors

"The soil type in the area is predominantly reddish lateritic soil with a silty loam topsoil texture". (what is the Taxonomic classification for this soil?)

Response: The soil taxonomy class is latosolic. The information has been included in the text

It would be beneficial to show the infiltration rate for this soil. 

Response: The information has been included in the text

In the statement below you are suggesting the adjustment of fertilization timing - can you be specific as to timing for practical purposes?

"Because fertilization before rainfall produces serious agricultural non-point source pollution, adjusting fertilization time reasonably could be an effective measure to avoid rainfall scouring after fertilization and then reduce nitrogen pollution in sugarcane areas."

Response: We are suggesting N-fertilization after rainfall to avoid rainfall scouring after fertilization and then reduce nitrogen pollution in sugarcane areas. The sentence before this has taken care of the recommendation. We have revised the section accordingly.

Comments on the Quality of English Language

 Some grammar/editorial corrections are needed throughout the manuscripts. 

Response: We have used Grammarly Premium software to edit and proofread the text. We have also given to a senior colleague who is vast in editing and proofreading to proofread.

Reviewer 3 Report

- abstract is too long, it should be more summarized.

Figure 2 is confusing, review scales and names.

The purpose was not clear in the introduction. Make it clear what problem the article seeks to solve.

Table 1 check formatting.

In the discussions, I miss a contextualization of the impact that the article can have in practice for producers.

Would it be a new recommendation? Make this point clear.

Author Response

We thank this Reviewer for his valuable to review our paper. We agree with the observations and comments to improve the work. The responses to the comments are highlighted below:

Comments and Suggestions for Authors

- abstract is too long, it should be more summarized.

 Response: The Abstract has been reduced, summarized

Figure 2 is confusing, review scales and names.

 Response: The Figure has been reviewed with respect to scale and naming

The purpose was not clear in the introduction. Make it clear what problem the article seeks to solve.

Response: The purpose has been clearly stated in the introduction

Table 1 check formatting.

  Response: Many thanks for this observation. The Table has been realigned. 

In the discussions, I miss a contextualization of the impact that the article can have in practice for producers.

Would it be a new recommendation? Make this point clear.

Response: We recommending reasonable adjustment of fertilization dose and post-rain application to reduce nitrogen fertilizer inflow into water bodies, avoid the intensification of eutrophication and nitrogen pollution in sugarcane producing areas in the region.

Reviewer 4 Report

editing - double spaces or missing spaces (line 55, 88, 114, 128, 129, maybe more...)

units - kg N ha-1 is not an SI unit - kg ha-1 N; sometimes, there are units without “.” (like mentioned kg N ha), sometimes you used “.” (like mg.L-1) - it should be unified

lines 163-166 - wrong format of text, probably because of added Figure

Table 1 - edit the table, so every number are underneath each other

The article is interesting in my opinion, although the sugarcane or monsoon climate is really my area of expertise. However, I present a question (or recommendation to further studies) to the authors about the methodology and observed results.

Firstly - How different is the weather in each year? Could the results be affected by the one-year experiment (2018)? Was the experiment conducted over multiple years?

Secondly - The fertiliser rates were chosen, I assume, according to the recommendations and practices for sugarcane. Other doses (higher, nowadays more of an effort to reduce) could also affect the results. Especially lower doses would probably lead to lower nitrogen losses. The question is whether sugarcane yields would be sufficient. Is this also the subject of your research?

I recommend a grammatical revision of the text

Author Response

We thank this Reviewer for his valuable to review our paper. We agree with the observations and comments to improve the work. The responses to the comments are highlighted below:

Comments and Suggestions for Authors

editing - double spaces or missing spaces (line 55, 88, 114, 128, 129, maybe more...)

Response: We have used Grammarly Premium software to edit and correct the spacing. We have read over again for further editing.

units - kg N ha-1 is not an SI unit - kg ha-1 N; sometimes, there are units without “.” (like mentioned kg N ha), sometimes you used “.” (like mg.L-1) - it should be unified

Response: All kg N ha-1 and kg P ha-1 have been written as kg.ha-1 N and kg.ha-1 P, respectively. All the units have been unified.

lines 163-166 - wrong format of text, probably because of added Figure

Response: The sentence has been revised

Table 1 - edit the table, so every number are underneath each other

 Response: Many thanks for this observation. The Table has been realigned. 

The article is interesting in my opinion, although the sugarcane or monsoon climate is really my area of expertise. However, I present a question (or recommendation to further studies) to the authors about the methodology and observed results.

Firstly - How different is the weather in each year? Could the results be affected by the one-year experiment (2018)? Was the experiment conducted over multiple years?

Response: The weather data cannot be the same in each year. Due to technical reasons, we could only conduct the experiment in 2018. The study can be repeated over time, in multiple years. This is one of our recommendations.

Secondly - The fertiliser rates were chosen, I assume, according to the recommendations and practices for sugarcane. Other doses (higher, nowadays more of an effort to reduce) could also affect the results. Especially lower doses would probably lead to lower nitrogen losses. The question is whether sugarcane yields would be sufficient. Is this also the subject of your research?

 Response: Yes, different doses could also affect the results. As mentioned, lower doses would probably lead to lower nitrogen losses, we didn’t verify this and may be sugarcane yields would not be sufficient, we don’t know. These are subjects for further research. These are parts of our recommendations.

Comments on the Quality of English Language

I recommend a grammatical revision of the text

Response: We have used Grammarly Premium software to edit and proofread the text. We have also given to a senior colleague who is vast in editing and proofreading to proofread.